# A mathematical framework for understanding the spontaneous emergence of complexity applicable to growing multicellular systems

**Lu Zhang[1,2], Gang Xue[3], Xiaolin Zhou[4], Jiandong Huang[5,6]\*, Zhiyuan Li[1,3]\***

**1** Center for Quantitative Biology, Academy for Advanced Interdisciplinary Studies, Peking University, Beijing, China, **2** Institute of Zoology, Chinese Academy of Sciences, Beijing, China, **3** Peking-Tsinghua Center for Life Sciences, Academy for Advanced Interdisciplinary Studies, Peking University, Beijing, China, **4** Tsinghua-Peking Center for Life Sciences, Tsinghua University, Beijing, China, **5** School of Biomedical Sciences, Li Ka Shing Faculty of Medicine, The University of Hong Kong, Pokfulam, Hong Kong SAR, China, **6** Chinese Academy of Sciences (CAS) Key Laboratory of Quantitative Engineering Biology, Shenzhen Institute of Synthetic Biology, Shenzhen Institutes of Advanced Technology, Chinese Academy of Sciences, Shenzhen, China

\* jdhuang@hku.hk (JH); zhiyuanli@pku.edu.cn (ZL)

**Data Availability Statement:** All relevant data are within the manuscript and its Supporting Information files.

## Abstract

In embryonic development and organogenesis, cells sharing identical genetic codes acquire diverse gene expression states in a highly reproducible spatial distribution, crucial for multicellular formation and quantifiable through positional information. To understand the spontaneous growth of complexity, we constructed a one-dimensional division-decision model, simulating the growth of cells with identical genetic networks from a single cell. Our findings highlight the pivotal role of cell division in providing positional cues, escorting the system toward states rich in information. Moreover, we pinpointed lateral inhibition as a critical mechanism translating spatial contacts into gene expression. Our model demonstrates that the spatial arrangement resulting from cell division, combined with cell lineages, imparts positional information, specifying multiple cell states with increased complexity—illustrated through examples in *C.elegans*. This study constitutes a foundational step in comprehending developmental intricacies, paving the way for future quantitative formulations to construct synthetic multicellular patterns.

## Author summary

Embryonic development shapes our bodies from a single cell, determining the placement of the head and tail. But how do cells, all sharing the same genetic code, precisely know what to become? Our mathematical model cracks this code. Envision your information being provided by your neighbors and your 'mom'—the division lineage. Then, appropriate regulatory networks (such as lateral inhibition, the most effective network motif) transform this information into diverse yet robust gene expressions. This math model helps us see the rules behind spontaneously growing complexity, guiding us to create new

**Funding:** This work was supported by the National Key Research and Development Program of China (No. 2021YFA0910700 to ZL), and National Natural Science Foundation of China (No. T2321001 to ZL). LZ was supported in part by the Postdoctoral Fellowship of Peking-Tsinghua Center for Life Sciences. The funders had no role in study design, data collection and analysis, decision to publish, or preparation of the manuscript.

**Competing interests:** The authors have declared that no competing interests exist.

patterns of cells. It's a big step in understanding how bodies form and opens doors to building cool new structures.

## Introduction

During embryonic development, a single fertilized egg transforms into a complex organism with diverse cell types arranged in a highly reproducible spatial pattern [1]. Despite being genetically identical, cells differentiate into distinct cell types with unique functions, and activate specific gene expressions consistently in certain positions and times, ensuring robustness of development [2]. On the one hand, the diversity in gene expression is essential for cells to carry out various functions in the embryo [3,4]; on the other hand, to function appropriately at each specific stage, cells must robustly enter the designated state [5–7]. Researchers are thus confronted with the question of what mechanisms enable the diversity and reproducibility of cell fate patterns during development, and how to assess possible mechanisms.

Over the last 50 years, the concept of positional information has become central to understanding the mechanisms that drive embryonic development. Positional information is used to describe the unique position of cells relative to one or more points in the developing system [8]. Wolpert postulated that cells acquire positional identities in a coordinate system and subsequently interpret their positions to develop in specific ways [9]. Crick first proposed gradients of morphogens as one of the main mechanisms for establishing positional information [10]. Since then, experiments on invertebrate and amphibian limb regeneration have supported the idea that cells acquire positional identity through the action of morphogens [11,12]. For example, in the newt limb, a membrane molecule called Prod1 is graded from one end to the other along the proximodistal axis, forming the molecular basis of positional information [13]. Similarly, during early *Drosophila* development, the maternal morphogen gradient provides positional information to direct gene expression patterning [14]. Through a combination of experiments and analysis, Bialek and his colleagues revealed how the primary morphogen gradient achieves precision and reproducibility in the early stages of *Drosophila* embryo pattern formation [15]. In their works, they used the concept of mutual information to quantify positional information, calculating it as the reduced Shannon entropy of cell states across the entire system given cells' positions in the unit of "bit". The network of four gap genes was shown to transfer positional information close to the upper bound of physical limit [16]. These studies focus on systems that extract positional information from established morphogen gradients. However, in a myriad of different organisms, this maternal morphogen-directed mechanism is either absent or insufficient during early embryogenesis. Therefore, it is crucial to explore how a developing system can acquire and amplify positional identity through alternative means.

Organized multicellularity can emerge spontaneously in organisms with limited or no external sources of information. For example, in the model organism *C. elegans*, cell fate transitions are primarily regulated by cell-cell communication without any known morphogen gradient [17–19]. In higher mammalians, spontaneous production of signaling waves and polarization are prominent in directing the embryonic development [20–22]. In these diverse species where external morphogens are unable to support the entire developmental process, autonomous polarity (like Par proteins and planar cell polarity) [23–25] and cell-cell signaling mechanisms (like ligand-bound receptors and Notch signaling) [26,27] appear to be critical. In these systems, the development process begins with a fertilized egg with zero bit of positional information. However, as cells divide and differentiate, the positional information of the entire

system increases without internal gradient sources. Therefore, it is essential to understand where this information comes from and how the system utilizes it.

Biophysical modeling approaches have long been employed to investigate spontaneous pattern formation. One such approach, Cellular Automata (CA), was proposed by John von Neumann in the early 1950s to imitate the self-replication capability of biological systems [28]. In this mechanism, a grid of "colored" cells evolves over several discrete time steps according to a set of rules based on neighboring cells' states, ultimately forming a specific pattern [29,30]. Another approach, the reaction-diffusion model, was introduced by Alan Turing in 1952, demonstrating how a reaction between two morphogens diffusing through tissue could result in self-regulating periodic biological patterns [31]. This mechanism has been used to replicate many natural patterns, such as zebrafish stripes [32,33] and seashell patterns [34], but experimental demonstrations of this mechanism in vivo are few and far between, and none identify both diffusible morphogens. The Clock and Wavefront model, proposed by Cooke and Zeeman in 1976, explains the formation of segments in the growing body axis of vertebrate embryos by a biological clock ticking at the posterior of the elongation embryo. The length of a developing segment is dictated by the distance a wavefront advances across the embryonic axis during a clock cycle [35]. While these mathematical models have merit, they have not yet been systematically applied to early development, which involves not only the emergence of patterns but also the reproducible expression of genes at each exact time and place [36]. While positional information might be used to quantify the diverse but precisely-positioned cell fates, understanding the process for the spontaneous creation of such information during early embryogenesis posed an intriguing challenge: how does the genetic code that is identical for all cells unfold into distinctive and reproducible spatial identities, and what process underlies the spontaneous creation of positional information during early embryogenesis [37]?

To investigate the underlying mechanisms of spontaneous complexity in embryogenesis, we developed an agent-based model that simulates the earliest stages of development by incorporating both fate decision and cell division. Each agent in the model represents a cell with an identical genetic network for cell state transition, and cells only communicate through local contact signaling, rather than diffusible morphogens. By using this division-decision model, we identified genetic networks that allow the growing system to achieve high positional information, beginning with a single zygotic cell. We uncovered lateral inhibition as a basic network motif for such spontaneous information genesis. Through further examination of these motifs, we were able to formulate how a growing system acquires positional information from the spatial-proximity and lineage-inheritance of cells, then used events during early development of *C.elegans* as the examples to illustrate these two sources of information. Our analysis of the noise-resistance and attractor landscape of the screened networks also highlights the role of cell division in providing positional cues and aiding the system to enter high-information states. Taken together, our research establishes a theoretical framework for the spontaneous emergence of complexity in early embryogenesis, providing computational insights into the comprehension and design of autonomous multicellularity.

## Results

### Lateral inhibition serves as a basic motif for information increase in the single-gene growing system

We began our investigation by studying the simplest division-decision system, in which a one-dimensional array is formed by consecutive divisions starting from a single cell. The binary expression states of a single gene A, $g_i^A(t) = 0$ or 1, characterize the state of the $i$-th cell as

$\vec{G}_i(t) = (g_i^A(t))$. The rules of gene expression updating were described in Eq 2 in the Method section.

Gene expression at time $t + 1$ is determined by a step function of the regulation-weighted summation of gene expressions at time $t$, with the threshold of the step function set to 0 [38][39]:

$$g_i^A(t+1) = \begin{cases} 1, & \left(a_{i,i}^{A,A}g_i^A(t) + \sum\limits_{k=i-1,i+1} a_{i,k}^{A,A}g_k^A(t)\right) > 0 \\[2mm] 0, & \left(a_{i,i}^{A,A}g_i^A(t) + \sum\limits_{k=i-1,i+1} a_{i,k}^{A,A}g_k^A(t)\right) < 0 \\[2mm] g_i^A(t), & \left(a_{i,i}^{A,A}g_i^A(t) + \sum\limits_{k=i-1,i+1} a_{i,k}^{A,A}g_k^A(t)\right) = 0 \end{cases} \qquad \text{(Eq 5)}$$

In describing the regulation of gene A within and between cells, the value of $a_{i,i}^{A,A}$ indicates the intracellular regulation of gene A towards itself, and the value of $a_{i-1,i}^{A,A}$ indicates the intercellular regulation of gene A in the $i$-th cell towards gene A in its neighboring cells $i$-$1$ (1 for activation, -1 for inhibition, and 0 for no interaction). As no left-or-right information was imposed on the system, $a_{i-1,i}^{A,A}$ is the same as $a_{i,i-1}^{A,A}$ for symmetry. The set $\{a_{i,i}^{A,A}, a_{i-1,i}^{A,A}\}$ defines all interactions in this system and is genetically encoded, therefore, we refer to this set as the "genetic network."

For one dimensional system, there are nine possible genetic networks (Figs 1B and S2A). As the gene expression is binary, the maximum positional information in this single-gene system is 1.0 bit. We simulated all nine genetic networks starting from a single cell to assess the maximum positional information achievable by the stable states. In each generation, the state of each cell was updated twenty times before simultaneous division of all cells in that generation.

We discovered that only one genetic network, O1, could achieve maximum positional information of 1.0 bit, for the initial cell state $\vec{G}_1(t_{initial}) = (g_1^A(t_{initial}) = 1)$. In contrast, the other eight genetic networks remained at 0.0 bit throughout division and decision (Fig 1C). In addition, all genetic networks maintained 0.0 bit positional information throughout development if the initial cell state is $\vec{G}_1(t_{initial}) = (g_1^A(t_{initial}) = 0)$. The O1 genetic network comprises one intracellular self-activation and one intercellular inhibition (Fig 1D), and an example of the gene expression pattern during system growth governed by the O1 was shown in Fig 1E. The development pattern of the system regulated by network O1 consists of three updates in each generation due to the fact that the system can reach stable states via three updates of gene expression in each generation. This self-activation/cross-cell-inhibition connection resembles the lateral inhibition regularly reported in multiple biological patterns [27,40,41], which prompted us to examine the mechanism of information genesis in greater depth.

We observed that the increase in information content in the O1-directed development is due to the update of the contact matrix $C$ induced by cell division. The positional information increases from 0.0 bit to 1.0 bit at step 8, in the 4-cell stage. As shown in Fig 1E, from generation 2 to generation 3, cell 2 and cell 3 divides into cell 4, 5 and cell 6, 7, respectively. These four daughter cells inherit cell states from cell 2 and cell 3, with gene A in the ON state for all four of them. Meanwhile, the action of a division changes the contact relationship between cells: cells 4 and 7 only have one neighbor with gene A ON, whereas cells 5 and 6 have two neighbors with gene A ON (Fig 1F). The contacting state ($CS$) differentiates four cells at step 7 into two groups (i.e., those contacting one $g^A = 1$ cell, vs. those contacting two $g^A = 1$ cells), providing the system with potential information of 1.0 bit. At step 8, regulated by the rules of genetic network O1, the cells contacting two neighbors with high A expressions are strongly

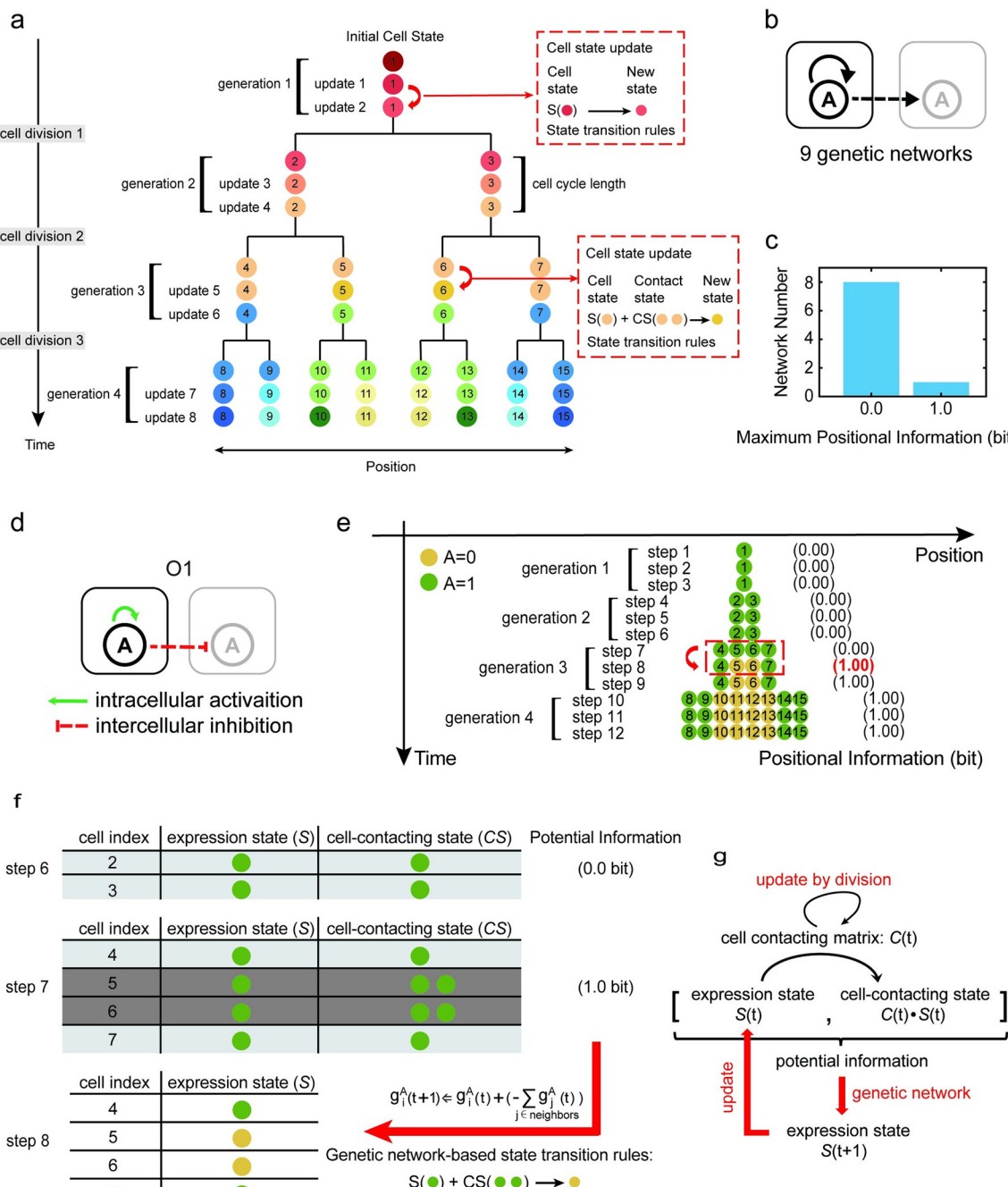

**Fig 1. One-dimensional division-decision growing model and its maximum information pattern in the single-gene system.** (a). A diagram illustrating our one-dimensional division-decision growing model. In our model, cells with the identical genome divide and differentiate from a single cell state to produce a multicellular system with several cell states. Distinct colors are used to indicate different cell states. (b). Possible intracellular and intercellular interactions in the single-gene system. There are nine possible types of genetic networks. The black arrow represents the possible interaction relationship, like an activation, an inhibition or an absence of interaction. Solid and dotted lines represent intracellular and intercellular interaction relationships, respectively. (c). A distribution diagram of the maximum positional information in the single-gene system. (d). The exclusive genetic network which controls maximum positional information pattern formation in the single-gene system. (e). The pattern of gene expression varies over time in a single-gene system regulated by the genetic network O1. At step 8, the system obtains 1.0 bit positional information. The time-course of positional information is calculated and shown in the right of the pattern. The binarized gene A expression level determines the cell state. Green circles represent gene A expressed while yellow circles represent gene A unexpressed. (f). Explanation of increased positional information in the single-gene system. The maximum positional information 1.0 bit is created by the new space network resulted from cell division. Cell 4, cell 5, cell 6, and cell 7 inherit cell states from their mothers when cell 2 and cell 3 divide. The

expression state matrix ($S$) and the cell-contacting state matrix ($CS$) comprised the concatenated matrix ([$S,C\cdot S$]), which changed after cell division. The new concatenated matrix contains 1.0 bit potential information which could potentially distinguish different two cell states. With the time moving forward, updated cell states according to the genetic network O1 in (d), cell 5 and cell 6 acquires new cell states. At same time, positional information in 4-cell stage increases to 1.0 bit. (g). A diagram illustrating the mechanics of information growth in division-decision systems. The cell contacting matrix ($C$), together with the current cell expression states ($S$), form the potential information source manifested in the concatenated matrix ([$S,C\cdot S$]). Appropriate genetic networks convert the potential information contained in the concatenated matrix ([$S,C\cdot S$]) into the updated expression states ($S$), resulting in an increase in the positional information. Cell division may change the cell contacting matrix ($C$), which therefore increases the potential information.

inhibited and turn off their gene A expressions. Therefore, cells 5 and 6 update their gene A expression states from ON to OFF. This transition results in four cells with two distinct cell states, so that the system's positional information reaches 1.0 bit. In other words, the genetic network O1 transforms 1.0 bit of potential information at step 7 into 1.0 bit positional information at step 8 (Fig 1F). In contrast, the other eight genetic networks could not efficiently transfer potential information into the system's actual positional information (S2A Fig). Usually, the process by which genetic networks convert potential information into positional information has some degeneracy. Multiple inputs in concatenated matrix [$S,C\cdot S$] might result in the same output in gene expression states $S$ (Fig 1G). For example, as shown in Fig 1E, from generation 3 to generation 4, the neighbors of cells 10 and 11 are in distinct states. Yet, the degeneracy prevented the genetic network from updating these two cells into two distinct cell states. Genetic networks exhibiting a smaller degree of degeneracy are more capable of converting information.

In addition to Eq 5, we also attempted to control gene expression using additional gene regulatory logics and updating functions [42–44]. We found another genetic network, O2 (Eq S1 and S3 Fig), of the similar topology with O1, has the same performance as O1 (details in S1 Appendix).

Taken together, in the single-gene system, the results above demonstrate that cell division can change the spatial contacting state of a system, offering potential information that can be transformed into positional information with the appropriate genetic network (Fig 1G).

## Spatial contact and cell lineage amplify potential information in a two-gene division-decision system

Next, we investigated the principles of information amplification in two-gene systems. With two genes A and B with binary ON and OFF states, a cell's state can be expressed as $\vec{G}_i(t) = \left(g_i^A(t), g_i^B(t)\right)$, and there are four possible states for each cell: none expressed $\left(g_i^A(t) = 0, g_i^B(t) = 0\right)$, both expressed $\left(g_i^A(t) = 1, g_i^B(t) = 1\right)$, and only one gene expressed $\left(g_i^A(t) = 1, g_i^B(t) = 0;\ or\ g_i^A(t) = 0, g_i^B(t) = 1\right)$. We used Eq 5 to update the gene expression of the cell. As mentioned in Method, the number of possible intracellular and intercellular regulatory networks (i.e., genetic networks) composed by two genes is 6561 (Fig 2A). Starting from four distinct initial states in a single cell, we simulated all 6561 genetic networks in the division-decision system, and recorded the maximal positional information each network could reach (Fig 2B).

With binary gene activities, the maximum positional information in the two-gene system is 2.0 bit. Our simulations revealed that only two genetic networks, named T1 and T1', are capable of driving systems to reach this maximal positional information at stable states (Figs 2C, 2D and S4) (S2 Appendix). Since systems can reach stable states via four updates of gene expression in each generation, development patterns of systems regulated by network T1 and T1' consist of four updates in each generation. Network T1 comprises four regulatory edges: intracellular self-activation of gene A, lateral inhibition between A genes across contacting

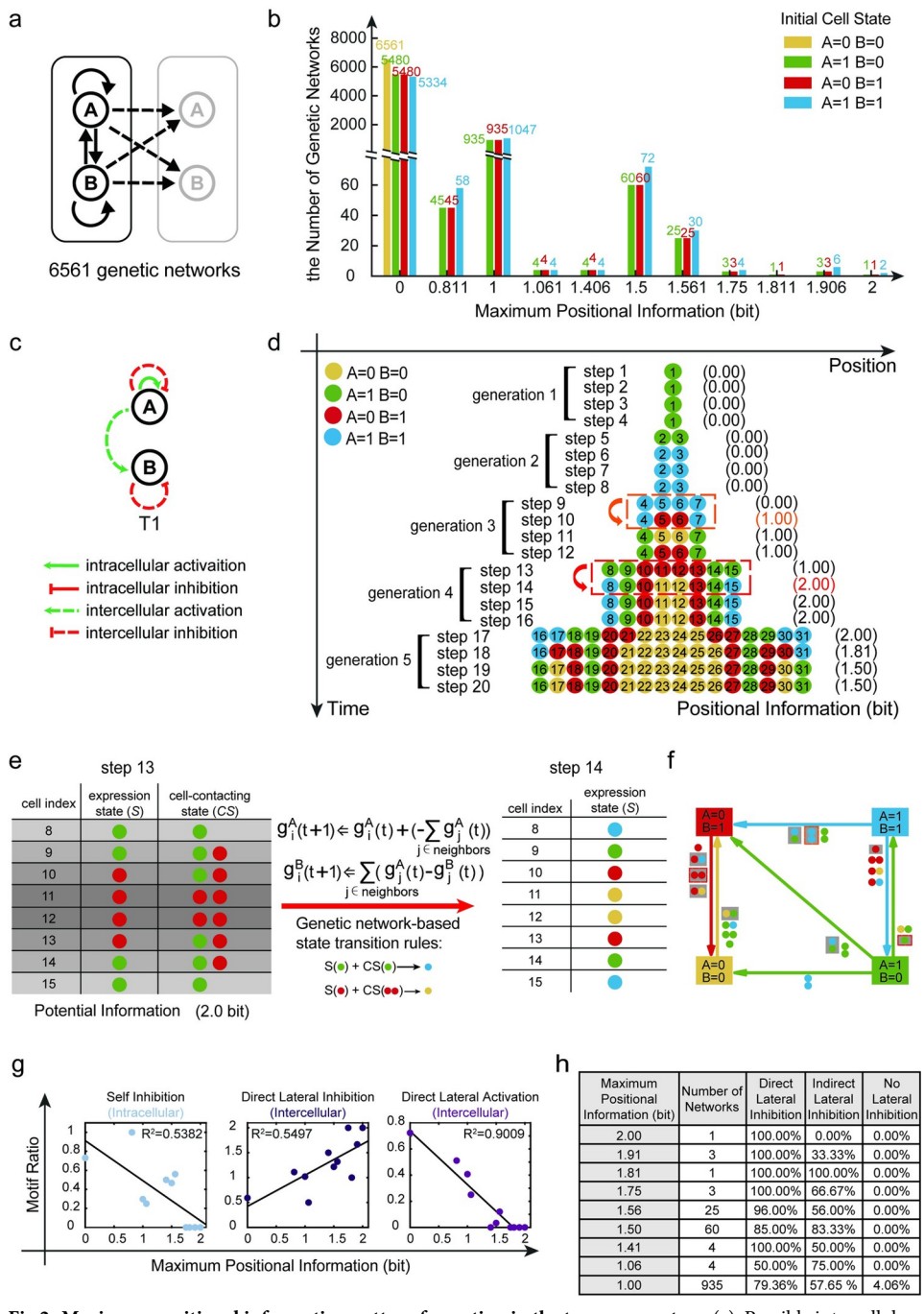

**Fig 2. Maximum positional information pattern formation in the two-gene system.** (a). Possible intracellular and intercellular interactions in the two-gene system. The number of possible genetic networks is 6561. The black arrow represents the possible interaction relationship, like an activation, an inhibition or an absence of interaction. Solid and dotted lines represent intracellular and intercellular interaction relationships, respectively. (b). The distribution diagram of the number of genetic networks corresponding to different maximum positional information under four different initial cell states. (c). The exclusive genetic network (T1) in the two-gene system that regulates maximum positional information pattern formation in a stable state. (d). The expression pattern of two genes throughout time and space. This pattern formation regulated by the genetic network T1 in (c). At step 14, system reaches 2.0 bit positional information. (e). Mechanisms for increasing positional information from 1.0 bit to 2.0 bit. When cells 4 to 7 divide, cells 8 to 15 inherit cell states from their mothers. At this time, the potential information in the concatenated matrix ($[S, C \cdot S]$) is 2.0 bit. With the time moving forward, updated cell states according to the genetic network T1 in (c), cell 8, cell 11, cell 12 and cell 15 alter their states. At step 14, positional information increased from 1.0 bit to 2.0 bit. (f). Transition linkages across four cell states under the regulation of genetic network T1. The number and color of the

circles next to the line indicate the number and state of the cell's neighbors, and an arrow points from the initial state of the cell to the final state of the cell. The similar cell state transition that took place in (d) is shown by circles with a gray background. (g). The relationship between maximum positional information and the proportion of times a regulatory motif occurs. The proportion of self-inhibition and direct-lateral-activation motif occurrence are negatively correlated with the maximum positional information. While the proportion of direct-lateral-inhibition motif occurrence is positively connected with the maximum positional information. (h). According to the maximum positional information achievable in the development system directed by genetic networks, networks with the same maximum positional information are grouped together. The first column represents the maximum positional information value for each genetic network group. The second column indicates the number of genetic networks in each group. The third column represents the ratio of direct lateral inhibition motif present in the group. The fourth column represents the ratio of indirect lateral inhibition motif present in the group. The fifth column represents the proportion of networks in the group that do not exhibit lateral inhibition motif.

cells, activation of gene B by gene A across contacting cells, and lateral inhibition between B genes across contacting cells (Fig 2C). The network T1' can be generated by swapping the A and B genes in the network T1 (S4A Fig). When the initial cell states are $\vec{G}_1(t_{initial}) = \left(g_1^A(t_{initial}) = 1, g_1^B(t_{initial}) = 0\right)$ or $\vec{G}_1(t_{initial}) = \left(g_1^A(t_{initial}) = 1, g_1^B(t_{initial}) = 1\right)$, systems regulated by network T1 reach multicellular stable states with 2.0 bit of positional information (Figs 2C, 2D and S4B). While when the initial cell states are $\vec{G}_1(t_{initial}) = \left(g_1^A(t_{initial}) = 0, g_1^B(t_{initial}) = 1\right)$ or $\vec{G}_1(t_{initial}) = \left(g_1^A(t_{initial}) = 1, g_1^B(t_{initial}) = 1\right)$, systems regulated by network T1' reach multi-cellular stable states with 2.0 bit positional information (S4A and S4C Fig). Due to the symmetry between networks T1 and T1', the ensuing analysis will only focus on network T1. Under the regulation of network T1, the positional information undergoes two leaps during the system's growth: the first leap occurs at step 10, as the system divides from the 2-cell stage into the 4-cell stage, increasing from 0.0 bit to 1.0 bit; the second leap occurs at step 14, as the system divides from the 4-cell stage into the 8-cell stage, increasing from 1.0 bit to 2.0 bit (Fig 2D).

We continued to investigate the mechanism behind the increase in information within the two-gene system. The first leap of information is similar to that of network O1. During the transition from generation 2 to generation 3, cells 2 and 3 divide into four identical cells (4 to 7), which exhibit the BLUE gene expression state ([A = 1,B = 1]). However, due to the number of neighboring cells, the four cells can be split into two categories. Specifically, cells 4 and 7 each contact one neighbor with high gene A, whereas cells 5 and 6 contact two neighbors with high gene A. By inhibiting gene A via lateral inhibition, cells 5 and 6 convert the potential information of 1.0 bit from the contacting state into positional information in the gene expression states (RED state, [A = 0,B = 1]).

The second leap of information is influenced not only by the contacting state *CS* but also by the current gene expression state *S*. From generation 3 to 4, cells 4 to 7 give birth to cells 8 to 15. Cells 8, 9, 14, and 15 inherit their mother cells' GREEN state (GREEN state, [A = 1,B = 0]), while cells 10 to 13 inherit the RED state (RED state, [A = 0,B = 1]). Immediately after division at step 13, the concatenated matrix ([*S*,*CS*]) is shown in Fig 2: Cells 8 and 9 share the same cell state (GREEN) but have different neighboring states (GREEN vs. GREEN and RED). On the other hand, cells 9 and 10 share identical neighboring states (GREEN and RED), but their cell states differ (GREEN vs. RED). As a result of these differences in neighboring and expression states, the eight rows of the concatenated matrix ([*S*,*CS*]) can be divided into four distinct classes with a potential information of 2.0 bit. Under the regulation of T1, at step 14, the states of cells 8, 15, 11, and 12 transition into BLUE and YELLOW. As a result, the system exhibits four distinct cell states with a positional information of 2.0 bit. This transition from two to four states demonstrates how the system converts the combined cell and neighboring states ([*S*, *CS*]) into updated positional information. As the differences between cells 9 and 10 are inherited from mother cells, this source of information come from cell lineage.

From the perspective of transition linkages across cell states [45], we observe that physical clustering of cells in the same state plays a crucial role in the emergence of new cell states during the development process in Fig 2D. As shown in Fig 2F, a BLUE cell ([A = 1,B = 1]) transitions to a RED state ([A = 0,B = 1]) when it encounters two neighboring cells with the same BLUE state, which occurs between steps 9 and 10 in Fig 2D. Similarly, a RED ([A = 0,B = 1]) cell transitions to a YELLOW state ([A = 0,B = 0]) when it contacts two cells in RED, which happens between steps 13 and 14. This cascade of new state emergence is facilitated by cell division, in which cells with identical states proliferate to promote the formation of the next new state.

In the two-gene system, we also simulated the growth of systems with different gene regulatory logics and updating functions. Using Eq S1 (S3A Fig), we discovered three genetic networks (T2, T3, and T4) that enabled controlled systems to attain multicellular stable states with 2.0 bit positional information (S5 Fig, See S1 Appendix for details).

## The lateral inhibition motif is central for the increase of positional information in the two-gene growing system

Among the 6561 genetic networks we examined, their capacity for increasing positional information during the division-decision process varied. To explore this, we investigated relationships with specific gene regulatory motifs, including direct positive feedback, mutual inhibition, negative feedback, and coherent and incoherent feedforward loops [39,46]. Focusing on six intracellular regulatory motifs and four intercellular regulatory motifs (S6A Fig), we considered only two initial cell states for simplicity. We grouped the genetic networks with the same maximum positional information after running for 5 generations. For each group of genetic networks, we calculated the "motif ratio" by dividing the number of each motif's occurrences by the total number of genetic networks in the group. We found two types of strong correlations between motif ratio and the maximum positional information (Figs 2G, S6B and S6C). Intercellular direct lateral inhibition positively correlated with maximum positional information, while intracellular self-inhibition and intercellular direct lateral activation negatively correlated (Figs 2G, S6B and S6C). Analyzing genetic networks categorized by their maximum positional information, we observed that direct or indirect lateral inhibition motifs were predominant in networks achieving positional information larger than 1.0 bit (Figs 2H and S6D). These results highlight the crucial role of intercellular lateral inhibition in generating high positional information patterns.

In actual biological processes, various embryonic development stages are initially guided by specific positional cues. In mimicking such initial asymmetry, we also examined the relationship between positional information and regulatory motifs in a one-dimensional multicellular model with one cell fixed at the left end of the system. The fixed cell is not allowed to divide nor change cell state, imposing an asymmetry on the system analogous to the P cell lineages in *C.elegans* development (S7A Fig). After evaluating all possible two-gene networks, we identified two capable of producing a four-striped pattern at the 9-cell stage, and fourteen capable of three distinct stripes (S7D Fig). Under a screened genetic network T5 (S7B Fig), the division-decision system produced a unique pattern of four stripes at the 9-cell stage (S7C Fig). While motif ratios showed no significant correlation with stripe number (S7E Fig), positional information negatively correlated with intercellular direct lateral activation and positively correlated with intercellular direct lateral inhibition motif ratios (S7F Fig). Moreover, all networks achieving positional information higher than 1.0 bit at the 9-cell stable state featured direct or indirect lateral inhibition motifs (S7G Fig). These findings underscore the pivotal role of lateral inhibition in amplifying positional information during development.

## Stable multicellular states with high positional information are ensured via cell division and pedigree

For accurate dynamic models in reality, a reasonable basin size is crucial for stable states at a biological level. The basin size represents the total initial cell states evolving into a stable state under specific genetic network regulation. For instance, in network T1, with two cells, there are 16 initial states, and only 10 represent stable multicellular states, suggesting a mean basin size of 1.6. Increasing the cell number to four results in 256 initial states, with a mean basin size of 3.9 (S8A and S8B Fig). Calculations for basin size and positional information at different cell numbers (regulated by T1) revealed a rapid increase in the largest basin size, reaching 30 for 4-cell states and 1275 for 8-cell states (Fig 3A and 3B). The high-information state, evolving through division-decision, shows a decreasing basin size ratio relative to the largest basin: 11 at the 4-cell stage and 71 at the 8-cell stage (Fig 3A and 3B). Basin size and positional information recordings for 4-cell and 8-cell stable states demonstrated that higher positional information correlated with smaller basin sizes (Fig 3C and 3D). Therefore, if the number of cells in the system remains constant, randomized initial states tend to evolve into a state with a large basin size, keeping positional information low. However, as the system grows from a single cell, early stages provide a larger basin ratio for the high-information state, "escorting" the system into its small basin through division. The pedigree procedure's cell division prevents the system from evolving into a stable state with limited positional information by eliminating numerous unstable starting states (Fig 3E).

Noisy cell state decision involves erroneous gene expression updates with a certain probability (Fig 3F), while differences in cell cycle length contribute to a divergence in the timing of cell divisions, leading to an asynchronous division pattern among cells within the same generation (Fig 3G). We simulate the division-decision system 10,000 times under these two distinct conditions for various genetic networks. Identifying the standard multicellular stable state corresponding to the maximum positional information (2.0 bit) in a noise-free and synchronous environment, we assess how likely systems can reach this state under conditions characterized by stochastic perturbations or asynchrony in cell division. The "accuracy" of the system, determined by the fraction of simulations reaching the standard state given gene expression variability and cell cycle length diversity, decreases for both forms. However, the impact differs slightly: in the case of the first condition, accuracy decreases as the cell cycle length increases, suggesting more mistakes with longer cell cycles. Conversely, for the second condition, where cell division is asynchronous, longer cell cycles enhance accuracy as they provide cells more time to rectify aberrant states. This pattern holds for other screened genetic networks using Eq S1, and T1-regulated systems exhibit better resistance to both conditions than other genetic networks (S8C and S8D Fig).

## The principle of increased positional information is applicable to continuous models

In our exploration of increasing positional information, we initially employed a basic Boolean model for gene expression updates. However, concerns about potential artificial outcomes prompted us to replace the Boolean model with an ordinary differential equation in this section [47,48]. This substitution aims to assess the continued applicability of the principle of increased positional information within a continuous model.

In the single-gene system, based on the network O1 (Fig 1D) and O2 (S3B Fig), we constructed different ordinary differential equations, by adopting AND or OR logics in combinatory regulations (Fig 4A and 4B, See Eqs S2-S3 in S1 Appendix). When the system reached the 4-cell stage (generation 3), we observed that the stable expression levels of gene A were

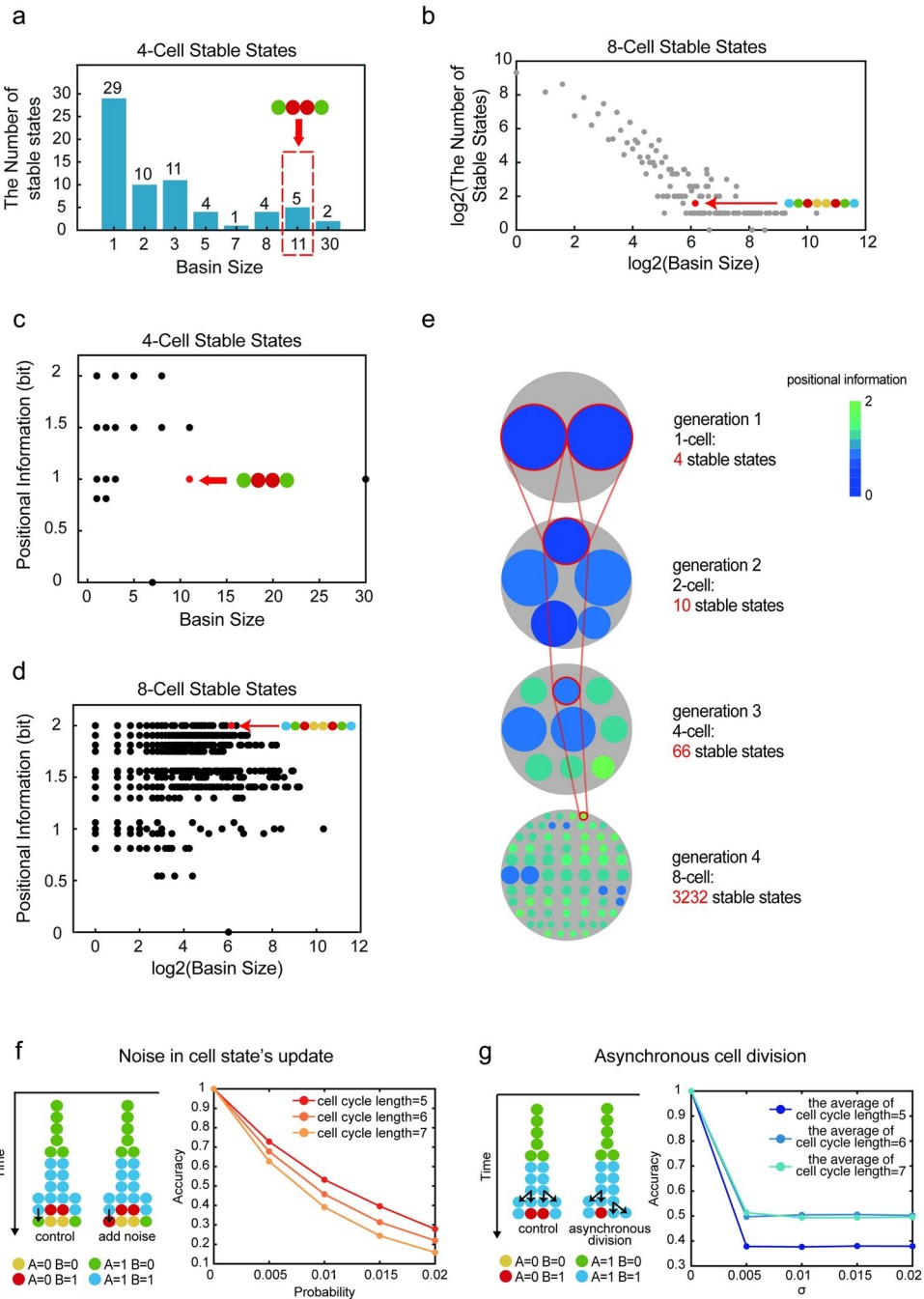

**Fig 3. Basin size and stability of multicellular stable states.** (a). The basin size of 4-cell stable states and the distribution diagram of the corresponding number of stable states under the regulation of genetic network T1. The red dotted box and red arrow denote the position of the 4-cell stable state formed by the growing system regulated by the genetic network T1. (b). In logarithmic coordinates, the basin size of 8-cell stable states and the distribution diagram of the corresponding number of stable states under the regulation of genetic network T1. The red arrow denotes the position of the 8-cell stable state formed by the growing system regulated by the genetic network T1. (c). Basin size and positional information corresponding to distinct 4-cell stable states under the regulation of genetic network T1. (d). In logarithmic coordinates, basin size and positional information corresponding to distinct 8-cell stable states under the regulation of genetic network T1. (e). Under the regulation of genetic network T1, multicellular stable states corresponding to various basin sizes were displayed in each generation in the schematic diagram. The areas of the smaller, colored circles represent the basin sizes of the distinct multicellular stable states, while the large gray circle represents the cumulative basin size of all stable states for each generation. And the circle's color corresponds to the positional information of the multicellular stable state. Red lines indicate stable states that appeared in the one-

dimensional division-decision growing system under the regulation of genetic network T1. (f). To make the cell state update with a certain probability of mistake, noise is introduced to the growing system. The noise that occurred during cell state update is depicted on the left panel. And the relationship between the probability of cell state update mistakes and pattern accuracy at various cell cycle lengths was illustrated in the right panel. (g). The left panel illustrates a representative instance of asynchronous cell division, while the right panel delineates the relationship between the standard deviation ($\sigma$) of cell cycle lengths and the accuracy of patterning under conditions of asynchronous division.

different across cells, indicating the increase of the positional information in the system and the embellishment of differential cell states (Fig 4A and 4B). Gene A expression, influenced by neighboring cells, led to varied stable states in cells with differing degrees of inhibition, consistent with Boolean model findings (S9A and S9C Fig). Moreover, transforming gene A's intracellular self-activation to constitutive expression aligned with the Boolean model's network O2 (Figs 4C, S3B and Eq S4). Despite lacking self-activation, the system still exhibited spontaneous pattern formation, highlighting the significance of the lateral inhibition motif (Fig 4B and 4C, see S1 Appendix for detail).

Analyzing three genetic networks under continuous expressions revealed that noise resistance varied (S1 Appendix). In generation 3, the system regulated by constitutive expression showed the highest noise resistance. In network O1, when intracellular self-activation and intercellular lateral inhibition are integrated through an AND-gate instead of an OR-gate, the pattern generated is more noise-resistant (Fig 4D). All networks experienced decreased noise resistance in generation 4 (S9F Fig).

We also applied the continuous model to simulate the two-gene growing system. By scanning the parameter space for network T1, we found that the stable expression pattern of genes A and B in generation 4 would match that of the Boolean model in the continuous model if gene A was AND-gate and gene B was AND-gate or OR-gate (S1 Appendix). The simulation results in pattern formation are consistent with the Boolean model (Figs 4E and S10). However, the continuous adaptation of the T1 model did not mirror the bi-stability pattern exhibited by gene B in the Boolean model, unless an intracellular self-activation was introduced (see S1 Appendix). This modified network, referred to as T6, showcased even more distinctive cell states in generation 4 (see S11D Fig). Nevertheless, when it comes to noise resistance, T1 demonstrated superior performance compared to T6 (Fig 4F).

## Mechanisms of the embryo increase its positional information in the early development of *C.elegans*

We sought to investigate the applicability of our concise mathematical framework to biological phenomena, particularly in relation to the development of *C. elegans*. This organism has long been recognized as a model for studying developmental processes due to its highly predictable lineage specifications that occur independently of external cues. To assess the accuracy of our framework, we utilized published experimental data from studies investigating cell-cell contact relationships and gene expression patterns during the early stages of *C. elegans* embryo development [18,49–53]. Using this available data, we digitalized the cellular states by binarizing their gene expressions, then performed calculations to determine the positional information and potential information at different time points during *C. elegans* embryo development, starting from the fertilized egg up until the 200-cell stage (Fig 5A, See S1 Appendix for details). Remarkably, our calculations revealed that the embryo possesses more than 5.0 bits of positional information by the time it reaches the 200-cell stage (See S1 Appendix for details). Notably, the positional information exhibits a clear and incremental trend, closely following the increases in potential information as development progresses (Fig 5A). This evident and

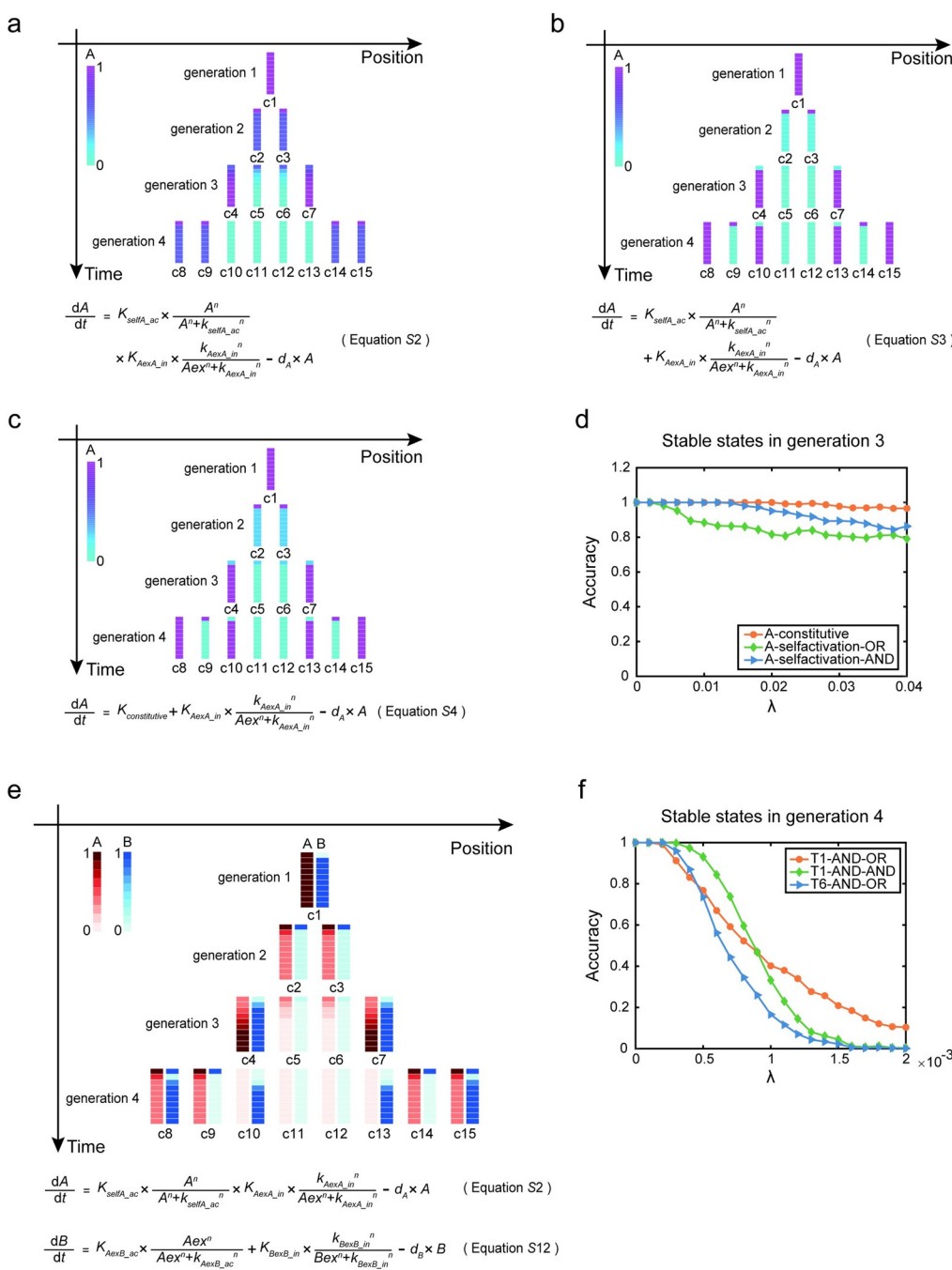

**Fig 4. The spatiotemporal expression patterns of genes in the continuous model and the noise resistance of particular patterns.** (a). The AND-gate regulatory function of the genetic network O1 in the continuous model and the spatiotemporal expression pattern of gene A simulated by this function. (b). The OR-gate regulatory function of the genetic network O1 in the continuous model and the spatiotemporal expression pattern of gene A simulated by this function. (c). The OR-gate regulatory function of the genetic network O2 in the continuous model and the spatiotemporal expression pattern of gene A simulated by this function. (d). Resistance to noise ($\lambda$) of patterns generated in generation 3 under various regulatory settings in single-gene systems. (e). The spatiotemporal expression patterns regulated by the genetic network T1 in the continuous model, and corresponding equations which regulate this pattern formation. (f). Resistance to noise ($\lambda$) of patterns generated in generation 4 under various regulatory settings in two-gene systems.

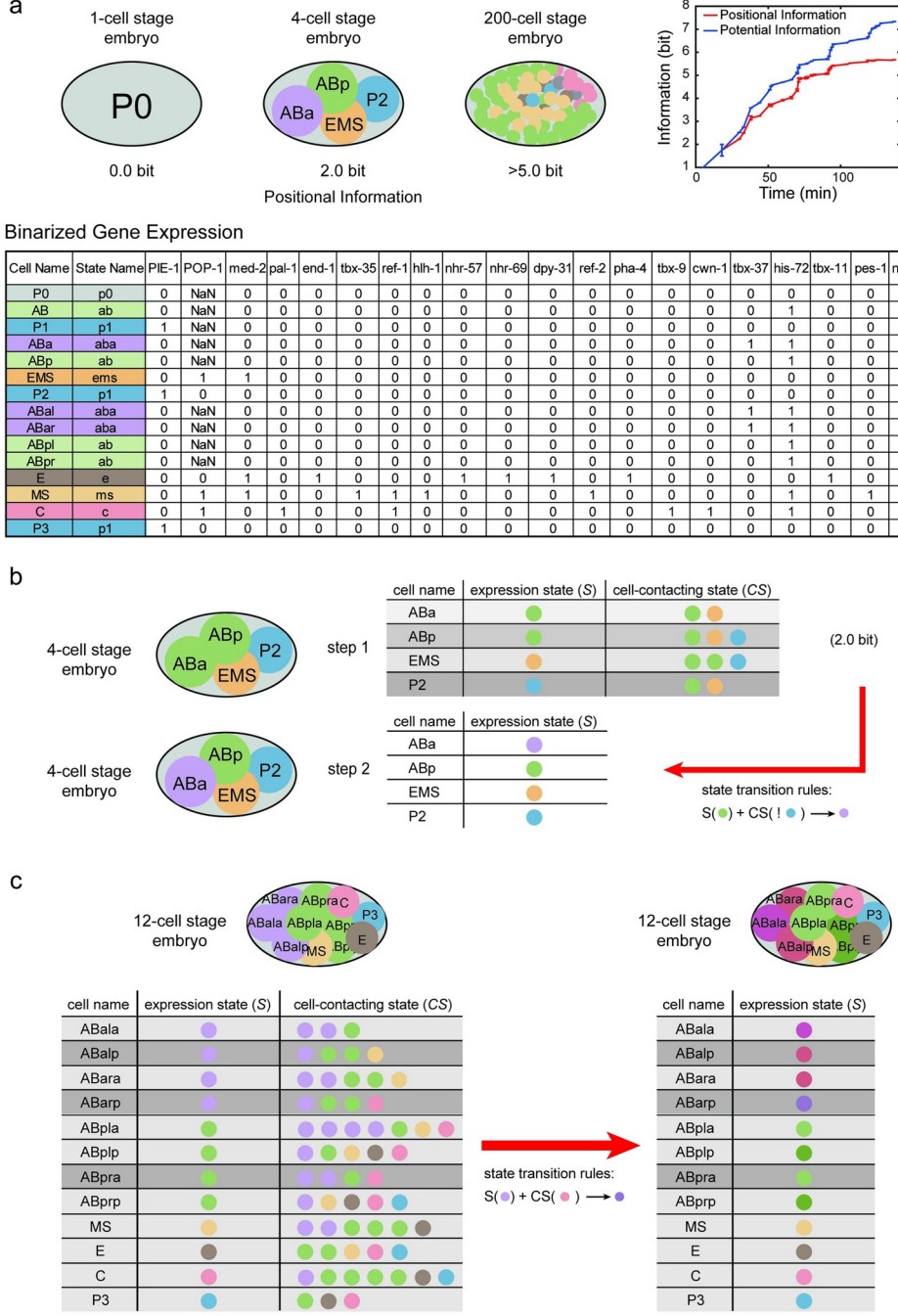

**Fig 5. Examples of positional information increase during *C.elegans* embryonic development.** (a). Positional information and potential information are estimated at each step of the development of the *C.elegans* embryo, from the fertilized egg to the 200-cell stage. The cell state is defined by the binarized gene expression profiles. Different states are marked by different colors, and named by the lower case of the first cell entering this state. (b). ABa cell transitions from the old cell state "ab" to the new cell state "aba" because it is not in contact with P2 cell during the 4-cell stage. (c). At the 12-cell stage, cell ABarp inherited the cell state "aba" and changed to "abarp" upon contact with C cell. Although the cell ABpra is also in contact with the C cell, its inherited cell state is "ab", so its cell state does not change.

consistent rise in positional information throughout development, without external sources, demands mechanistic explanations.

During the incipient stages of *C. elegans* embryo development, evidence suggests that neighboring cells provide cells with potential information (Fig 5B). Also, the significance of asymmetric cell divisions cannot be ignored, particularly in how cell polarity established prior to division. Our model integrates these elements by accounting for the inherent cell polarity and examining the influence of intercellular interactions on fate determination post-division. Specifically, at the 4-cell stage, cells ABa and ABp, progenies of progenitor cell AB, inherit the same "ab" cell state and equally express the Notch receptor. The anteroposterior disposition of ABa and ABp is contingent upon the polarity that progenitor cell AB establishes before division. Concurrently, the caudal cell P2 expresses the Delta ligand. Prior to the signaling event, the four-cell system exhibits three cell states, representing 1.5 bit of positional information. Meanwhile, cell P2 interacts with cell ABp, but not with cell ABa. Therefore, in a concatenated matrix that includes the states of each cell and their neighboring cells, there is 2.0 bit of potential information. During the signaling event, the Notch-Delta signal pathway prevents cell ABp from activating a specific set of genes associated with the "aba" state. As a result of lacking contact with cell P2, cell ABa transitions from the "ab" cell state to the "aba" cell state. Subsequently, all four cells demonstrate different gene expression states, resulting in the embryo possessing 2.0 bit of positional information. Hence, the spatial contact between cells conveys potential information regarding this event. This also underscores the interplay between intrinsic asymmetric divisions and extrinsic contact-mediated signaling in the developmental process.

In the 12-cell stage of *C. elegans* development, a noteworthy example involves the cells ABala, ABalp, ABara, and ABarp, all inheriting the common cell state "aba" from the ABal cell. The subsequent divergence in cell fate arises from the distinct interactions each cell has: ABalp interacts with MS, while ABarp interacts with the cell C. Consequently, ABalp adopts the "abalp" cell state, while ABarp transitions into the "abarp" cell state (depicted in Fig 5C). This process exemplifies the acquisition of positional information by the difference in neighboring cell states.

Simultaneously, another fate specification demonstrates how lineage inheritance contributes to the system's acquisition of positional information. Descendants from ABar, including cells ABpla, ABplp, ABpra, and ABprp, all share the "ab" cell state due to their lineage connection with ABar. While cells inheriting the "aba" cell state transform into the "abarp" cell state upon contacting cell C, as exemplified by ABarp, those inheriting the "ab" state remain unchanged even in contact with cell C, such as cell ABpra (Fig 5C). In this scenario, the positional information distinguishing ABarp and ABpra arises not only from the state of neighboring cells but also from the inherent state of the cells themselves, inherited through their maternal lineages.

## Discussion

How cells "know" their fates has long been a mystery in the field of embryonic development. To form a viable multicellular organism, organisms not only need to spontaneously generate patterns with complex local order, but also need to reproducibly form the same pattern in each embryogenesis process with little or even no external clues. In this study, we employ a one-dimensional division-decision model to investigate the mechanism by which a growing system acquires and amplifies positional information. Under this framework, the mechanism of information growth becomes clear. The cellular contacts $C$, together with the current expression states $S$, form the potential information source manifested in the concatenated matrix $[S, C \cdot S]$. Diversity in expression states and the number and state of nearby cells jointly promote the

variety of the concatenated matrix, resulting in a higher potential information than the positional information currently contained in $S$. Appropriate genetic networks translate the potential information contained in the concatenated matrix ($[S,C\cdot S]$) into the updated expression states, resulting in an increase in the positional information. These findings provide a novel and intuitive framework for understanding the genesis of information, and offer new insights into the process of embryonic development.

Our investigation emphasizes the critical importance of system growth in the spontaneous emergence of complexity in embryonic development. Cell division plays a vital role in this process by having multifold effects on the system. Firstly, cell growth increases the number of cells in the system, providing the basic material for development. Secondly, each division preserves "histories" of the system via pedigreed inheritance from mother to daughter cells, enabling cells to make different fate decisions in response to the same external signals. Furthermore, cell division alters the spatial contact of the daughter cells and provides the system with additional spatial clues, such as border and inside. For example, in our one-dimensional division-decision system, the first information leap occurs at the start of the 4-cell stage, where the division from 2 cells into 4 cells begins to set the difference in neighboring states. Such distinctions in "border" and "inside" spontaneously emerge as the system grows, allowing appropriate genetic networks to perform "edge detection," which is one strategy in synthetic biology to establish the initial heterogeneity of cellular states [54,55].

The importance of cell division in pattern formation has been noticed by previous researchers. S. Kondo and Rihito Asai discovered as early as 1995 that the strip patterns of *Pomacanthus* vary as the skin grows [56]. David G. Míguez et al. and Christopher Konow et al. found that in a system with continuous growth in one direction or radial direction, the orientation of stripes is associated with the growth velocity at the system boundary [57,58]. E. Crampin et al. examined pattern formation under various growth settings and demonstrated that the pattern's robustness was affected by the forms of growth [59,60]. However, when basin size is taken into consideration, we observed that cell division acts as a "funnel" to escort the system into high-information states. These high-information states themselves are highly unlikely in a system with a constant size due to their small basin size. In future research, exploring the role of cell division and system growth in other developmental processes and organisms would be a fascinating direction to pursue.

Appropriate genetic networks convert the potential information into explicit spatial information, realizing a cascade of information amplification. Our study has identified lateral inhibition as the basic network motif for this spontaneous information genesis under local signaling. In biological systems, lateral inhibition has been suggested to play a crucial role in pattern formation by establishing boundaries between different regions of the developing organism [61–63]. For instance, it is well established that the Notch signaling system performs lateral inhibition and is critical for early development [64,65]. In the salt-and-pepper pattern, lateral inhibition of the Delta and Notch genes in the Notch signaling pathway mediates the establishment of a boundary between neighboring cells [66]. In such a salt-and-pepper system, each cell tends to have a different cell state than its neighbors [27,67]. The Delta and Notch regulatory network can be simplified to our previously screened genetic network O1 in the single-gene system. Besides the ligand Delta, the ligand Jagged of one cell also interacts with the Notch receptor of its neighboring cell, causing the Notch Intracellular Domain (NICD) to be released. However, the released Notch Intracellular Domain represses Delta while activating Jagged. As a result, the intercellular inhibition network can be established by Notch and Delta, whereas the intercellular activation network can be produced by Notch and Jagged. Previous research found that increasing Delta production resulted in the formation of alternative cell states pattern, while increasing Jagged production caused all cells to have the identical cell state

[40,68]. These results are consistent with our findings that intercellular lateral inhibition regulates cells to have different cell states, thereby increasing the system's positional information, while intercellular lateral activation regulates all cells to reach the same cell state, thereby decreasing the system's positional information.

There is still much to be explored within our basic mathematical framework. The current framework considers that cell-cell signaling in space and inheritance of mother-cell states over time provide potential information, which serves as the foundation for cells to differentially update their gene expressions. However, it is also known that a cell's expression state can influence its movement, division orientation, and cell cycle duration, thereby forming a feedback loop of information amplification [69–71]. Furthermore, the mechanisms by which gene expression status affects division and location are also encoded in the genome [24,72]. Thus, future investigations should focus on how potential information originates from an all-cell-identical genome encoding, to fully understand the spontaneous emergence of complexity during embryonic development.

On the other hand, our current findings indicate that noise-induced fluctuations in gene expression levels can trigger state transitions in cells, reducing the positional information in multicellular stable states within the one-dimensional systems. From this perspective, since multicellular stable states with high positional information have smaller basin sizes, noise can cause the system to escape these states and develop towards others with lower positional information, which is disadvantageous for the system. However, previous studies have confirmed that noise can have not only negative effects on biological systems but also positive ones. For instance, during embryonic development, gene expression noise can enhance the robustness of organization by promoting cellular plasticity, allowing the embryo to correct organizational errors induced by dynamic cellular movements [73]. Moreover, noise in *hoxb1a*/*krox20* expression has been shown to promote the sharpening of boundaries between adjacent segments [74]. Therefore, in future research, we are also interested in exploring the potential beneficial effects of noise on positional information.

In conclusion, our research contributes a valuable quantitative framework for understanding the spontaneous emergence of complexity during early embryogenesis, which has implications not only in developmental biology but also in the field of synthetic biology. Our framework serves as a solid foundation and offers promising avenues for future research. It provides a valuable tool for quantitative analysis and optimization in the design of self-developing multicellular systems, both in fundamental research and industrial applications [75–78]. By establishing a quantifiable framework for understanding how complexity grows, researchers can define optimization functions for rational design. Firstly, it allows for computational exploration of "optimal" state-decision networks that maximize the transfer of potential information to positional information, taking into account lineage and spatial proximity relationships. Secondly, given a specific state-decision network, computational derivation of spatial arrangements capable of generating complex patterns becomes possible. Therefore, our framework serves as a blueprint for the rational design of complex patterns in synthetic biology, facilitating a deeper understanding of the mechanisms underlying the spontaneous generation of complexity.

While our research offers an initial step towards understanding the spontaneous growth of complexity during embryogenesis, it is important to recognize its limitations. One of these limitations is that our mathematical model only considers cell distribution in one dimension. In the real world, embryogenesis occurs in two or three dimensions, where the position of cells in space could have a greater impact on the system's positional information [17,26,79]. Due to the one-dimensionality of the system, for instance, there are only three conceivable neighbor configurations: no neighbors, one neighbor, and two neighbors. As a consequence, cells in the

system may only be able to recognize the system's edges and central region. The spatial arrangement of cells in a two-dimensional or three-dimensional system, however, may provide more potential information to the system because the number of neighbors of a cell can be significantly increased in these dimensions and there are numerous options for how to arrange the nearest neighbors. Future research should consider expanding the current model to include more dimensions to better understand the principles of gaining positional information. Another limitation of our model is that it only includes local signaling between nearest neighbor cells, while in biological systems, long-range regulation based on diffusible molecules is also important [80–82]. To account for this, future research should consider the effect of combined short- and long-range regulations on the system's positional information. Additionally, in our simulation, we focused on the scenario in which cells divide after reaching a stable state. However, in reality, cells do not always reach stable states within one cell cycle, and same-generational cells do not always divide simultaneously [53,83–85]. Further research should investigate the relationship between the duration of the cell cycle and the time it takes for cells to reach a stable state, the order in which cells divide within a generation, and how these factors impact pattern formation.

## Method

### Dynamical model for the one-dimensional division-decision system

Through gene-regulatory networks and cell-cell signaling, genes within a cell or between neighboring cells can activate or inhibit each other. These interactions between genes are genetically encoded, and we refer to each set of interactions as a "genetic network."

We developed a one-dimensional multicellular model in which all cells were derived from a single ancestor cell and shared identical genetic codes for cell state decisions (Fig 1A). The gene expression profiles of each cell were influenced by the states of its neighboring cells. Additionally, cell division occurred over time, and daughter cells inherited the expression states of their mothers. The system was designed to simulate the emergence of spatial patterns in one-dimensional space as the system grows.

This dynamical model is based on the Markov process, and the following rules are used to change the gene states and cell arrays:

**1. Update of the gene expression states by the genetic network.**   Genes and cells can interact with or regulate each other, i.e., activation, inhibition, etc. in many different ways, which can be encoded genetically. We define each set of interactions of genes and cells a "genetic network".

The state of the $i$-th cell at time $t$ was characterized by its gene expression profile $\vec{G}_i(t) = \left(g_i^1(t), g_i^2(t), \ldots, g_i^n(t)\right)$ ($n$ denotes the number of genes in the cell). $\vec{G}_i$ could be composed by one or multiple genes, depending on the model assumptions. Changes in $\vec{G}_i(t)$ were determined by the intracellular interactions quantified by the genetic network, as well as the intercellular interactions with neighboring cells (cell $i+1$ and $i-1$) quantified by the cell-cell signaling based on the identical genetic network:

$$\vec{G}_i(t+1) = f\left(\vec{G}_i(t), \vec{G}_{neighbors}(t)\right) \qquad \text{(Eq 1)}$$

In the first step, for simplicity, gene activity was set to binary form (1 for ON and 0 for OFF states), and the effect of genes on the dynamics takes the form of weighted summation.

Specifically, Eq 1 turns into the following form:

$$g_i^p(t+1)$$
$$= f\left(\sum_{q=1}^{n} a_{i,i}^{p,q} g_i^q(t), \sum_{q=1}^{n} a_{i,i-1}^{p,q} g_{i-1}^q(t), \sum_{q=1}^{n} a_{i,i+1}^{p,q} g_{i+1}^q(t)\right) \qquad \text{(Eq 2)}$$

$(i = 1, 2, \ldots, m; p = 1, 2, \ldots, n)$

In Eq 2, $m$ denotes the number of cells; $n$ indicates the number of genes involved in determining cell state and cell-cell signaling. The parameter $a_{i,i}^{p,q}$ weights the regulation from the $q$-th gene in the $i$-th cell to the $p$-th gene in the $i$-th same cell. This regulation, which depends on the genetic network, could be an activation interaction ($a_{i,i}^{p,q}>0$), an inhibition interaction ($a_{i,i}^{p,q} < 0$), or an absence of interaction between the $q$ and $p$ genes ($a_{i,i}^{p,q} = 0$)[38]. The first term in the right side of the bracket ($\sum_{q=1}^{n} a_{i,i}^{p,q} g_i^q(t)$) depicts how genes within the $i$-th cell regulate each other, and $a_{i,i}^{p,q}$ represents the genetic network within a cell. The second and third terms in the right side of the bracket depicts how the left ($i-1$) and right ($i + 1$) neighbors influence the gene expressions in cell $i$, and $a_{i,i-1}^{p,q}$ (or $a_{i,i+1}^{p,q}$) represents cell-cell signaling based on the genetic network between adjacent cells. Together, the genetic network ($a_{i,i}^{p,q}$ and $a_{i,i\pm1}^{p,q}$) is identical for all cells, and controls the dynamics of $\vec{G}_i(t)$.

The status of the whole system composed of $m$ cells, $S$, is composed by the expression states of all cells: $S = \begin{pmatrix} \vec{G}_1 \\ \ldots \\ \vec{G}_m \end{pmatrix}$, in the form of a $m \times n$ matrix. Without a morphogen gradient, only cells in direct contact can signal to each other, and we use the $m \times m$ cell-contacting matrix $C$ to represent this adjacency relationship. The signaling effect directly perceived by each cell, is the product of the expression states and the contact relationship, $C \cdot S$. Therefore, by Eqs 1 and 2, the updating of the system status by cell state changes can be abbreviated as:

$$S(t + 1) = F(A \cdot [S(t), C(t) \cdot S(t)]) \qquad \text{(Eq 3)}$$

In this equation, A represents the $a_{i,i}^{p,q}$ and $a_{i,i\pm1}^{p,q}$. It is an abstraction of the genetic network that is identical for all cells in the system. The concatenated input matrix, $[S, C \cdot S]$, combines the expression profiles of each cell with their contact relationships, from which each cell can updates its expression state.

**2. Update of the system size by cell division.** As growth continued, cells in the system divided once every cell cycle length (termed a "generation"), inheriting the states of their mother cells. For example, after one generation, the initial cell ($i = 1$), was divided into two daughter cells ($i = 2$, $i = 3$), and each daughter cell inherited the mother cell's current gene expression profile: $\vec{G}_2(t_{division}) = \vec{G}_3(t_{division}) = \vec{G}_1(t_{division})$. One of the two newly generated cells occupied the original mother cell's position, while the other randomly selected a position to the left or right of its sibling cell, then other existing cells on the left or right side would be shifted to far positions. If there were no cells on the left or right, this direction would be occupied preferentially. Therefore, as cells divide, the states of cell-contact signaling would change. The cell contact matrix $C$ updates after every cell division. Each generation consists of multiple updates of the gene expression states, as determined by Eq 2, to ensure the system reaches a stable steady-state or stable oscillation under the current number of cells. To ensure that cells in each generation reached stable states during genetic networks screening procedure, gene expression in each generation was updated 20 times throughout the simulation. To prevent the

system from introducing extra information, we also assumed that all cells of the same generation divide simultaneously.

## Enumeration of single-gene and two-gene networks

We enumerated genetic networks consisting of one and two genes. In the single-gene network, the state of a cell was determined by only one gene within the cell: $\vec{G}_i(t) = \left(g_i^1(t)\right)$. We considered all possible one-gene regulations, including all possible intracellular and intercellular interactions, as described by each of its two edges (Fig 1B). There is one edge in the intracellular interaction and one edge between genes in neighboring cells. In this binary model, each of these edges can be an activation interaction (1), an inhibition interaction (-1), or an absence of interaction (0) between the two genes. There are therefore $3^2 = 9$ possible types of genetic networks (S2A Fig). Based on the aforementioned division-decision model, we simulated this dynamic system starting from a single cell under all possible one-gene networks.

In the two-gene system, the state of one cell was determined by the expression profile of two genes in the cell: $\vec{G}_i(t) = \left(g_i^1(t), g_i^2(t)\right)$. We also considered all possible two-gene networks (Fig 2A). There are eight edges among four genes in two adjacent cells, and each edge can indicate an activation (1), inhibition (-1), or absence (0) of interaction between the two genes it is associated with. These forms $3^8 = 6561$ possible genetic networks. Similarly, we simulated this two-gene division-decision system under all possible genetic networks.

## Evaluation of the reproducible diversity by positional information

At each step of the system's growth, we calculated the positional information corresponding to the current cell states $\vec{G}_i(t) = \left(g_i^1(t), g_i^2(t), \ldots, g_i^n(t)\right)$ ($i = 1, 2, \ldots, m$; $m$ represents the current number of cells in the system). Positional information ($PI(state, position)$), defined as the reduced uncertainty of gene expression states given the cellular position, quantifies both diversity and reproducibility [16].

$$PI(state,\ position) = H(state) - H(state \mid position) \tag{Eq 4}$$

In Eq 4, "state" refers to the gene expression profile of the cell, denoted by $\vec{G}_i(t) = \left(g_i^1(t), g_i^2(t), \ldots, g_i^n(t)\right)$, while "position" refers to the spatial location of the cell. $H(state)$ term represents the diversity of cell states in the system. Shannon entropy is used to measure the uncertainty or randomness of a probability distribution. In this case, $H(state)$ is calculated based on the states of $m$ cells, where each cell has a gene expression profile denoted by $\vec{G}_i(t) = \left(g_i^1(t), g_i^2(t), \ldots, g_i^n(t)\right)$. $H(state|position)$ term represents the conditional entropy of the gene expression states given the cellular position. Conditional entropy measures the remaining uncertainty of a random variable (cell state) given the knowledge of another random variable (position). In this context, $H(state|position)$ captures how much uncertainty in cell states can be reduced by knowing the spatial location of the cells. By subtracting $H(state|position)$ from $H(state)$, we obtain the positional information ($PI$) for the current system. $PI(state, position)$ quantifies the reduced uncertainty or increased information about cell states when considering the cellular position. It represents both the diversity and reproducibility aspects of the system.

S1A Fig depicts two scenarios of calculating positional information for a simple one-dimensional system with 24 cells, each having 4 potential states. To measure the uncertainty for the two systems we assess the frequency of each state and calculate the total entropy ($H(state)$ = 2.00 bit for the left system and 1.98 bit for the right systems). In the left system, since the gene expression levels within cells are rigidly governed by regulatory rules without any variance

from internal or external noise, cell states are entirely determined by their positions, resulting in a conditional entropy of 0 bit. Conversely, the right system incorporates variability through both regulatory rules and stochastic influences, manifesting in reduced consistency of cell states at identical positions and a non-zero conditional entropy of 0.68 bit. The positional information for each system is then determined by subtracting the conditional entropy from the total entropy. Consequently, the first system has 2.0 bit of positional information, while the second system has 1.32 bit.

By considering the positional information, we can quantify how much uncertainty in cell states is reduced given the cellular position. This reduction in uncertainty indicates that certain states are more reproducible or restricted to specific spatial locations, leading to an increase in positional information.

In addition to positional information, we also defined the concept of "potential information". At each step of the system update, the concatenated input matrix $[S, C \cdot S]$ in Eq 3 provides each cell with the cell-specific input for cell state decisions. Therefore, the mutual information between $[S, C \cdot S]$ and cellular position is defined as "potential information", as it offers potential differences between cells. This potential information needs to be processed by the identical genetic network $A$ to finally update the expression profiles.

## Supporting information

**S1 Appendix. Detailed description of the division-decision model and the experimental data set.**
(DOCX)

**S2 Appendix. Table about the genetic networks of *C.elegans* with positional information.**
(XLSX)

**S3 Appendix. Table about the booleanization of *C.elegans* gene expressions.**
(XLSX)

**S4 Appendix. Table about the cell contact relationships in *C.elegans*.**
(XLSX)

**S5 Appendix. Related code and data for this manuscript.**
(ZIP)

**S1 Fig. An example of how positional information is calculated.** (a). Shannon entropy of cell states minus conditional entropy of cell states equals positional information. The conditional entropy is low (left table) if the cell state (cell states are displayed in different colors) is defined according to the position in several samples; if the cell state is still uncertain when the position is given, the conditional entropy is large (right table).
(TIF)

**S2 Fig. Multicellular stable states of four cells regulated by nine genetic networks.** (a). Starting with high levels of gene A expression in neighboring four cells, systems can achieve numerous multicellular stable states under the regulation of different genetic networks when the number of cells in the system is kept constant at four
(TIF)

**S3 Fig. The modified gene expression update function regulates the generation of maximum positional information patterns in the single-gene system.** (a). Modified gene expression update function. This modified function determines the gene expression state in the next time step based on the sum of all associated gene expression states in the current time step plus

one. (b). The genetic network which controls maximum positional information pattern formation in the single-gene system. (c). The pattern of gene expression varies over time in a single-gene system regulated by the genetic network O2. At step 8, the system obtains 1.0 bit positional information. The time-course of positional information is calculated and shown in the right of the pattern. The binarized gene A expression level determines the cell state. Green circles represent gene A expressed while yellow circles represent gene A unexpressed. (d). The relationship between positional information and time in the growing system regulated by the genetic network O2. (e). Mechanisms of increased positional information in the single-gene system. The maximum positional information 1.0 bit is created by the new space network resulted from cell division. Cell 4, cell 5, cell 6, and cell 7 inherit cell states from their mothers when cell 2 and cell 3 divide. The expression state matrix ($S$) and the cell-contacting state matrix ($CS$) comprised the concatenated matrix ($[S, C \cdot S]$), which changed after cell division. The new concatenated matrix contains 1.0 bit potential information which could potentially distinguish different two cell states. With the time moving forward, updated cell states according to the genetic network O2 in (b), cell 5 and cell 6 acquires new cell states. At same time, positional information in 4-cell stage increases to 1.0 bit.
(TIF)

**S4 Fig. Under distinct starting cell state settings, networks T1 and T1' regulate the spatial and temporal pattern of two genes' expression.** (a). Genetic network T1' in the two-gene system that regulates maximum positional information pattern formation in a stable state starting with the cell state $\vec{G}_1(t_{initial}) = \left( g_1^A(t_{initial}) = 0, g_1^B(t_{initial}) = 1 \right)$. At step 14, system reaches 2.0 bit positional information. (b). Genetic network T1 in the two-gene system that regulates maximum positional information pattern formation in a stable state starting with the cell state $\vec{G}_1(t_{initial}) = \left( g_1^A(t_{initial}) = 1, g_1^B(t_{initial}) = 1 \right)$. At step 14, system reaches 2.0 bit positional information. (c). Genetic network T1' in the two-gene system that regulates maximum positional information pattern formation in a stable state starting with the cell state $\vec{G}_1(t_{initial}) = \left( g_1^A(t_{initial}) = 1, g_1^B(t_{initial}) = 1 \right)$. At step 14, system reaches 2.0 bit positional information.
(TIF)

**S5 Fig. The modified gene expression update function regulates the generation of maximum positional information patterns in the two-gene system.** The diagram of transition linkages among four cell states, the relationship between positional information and time, and the time-space pattern of gene expression are depicted under the regulation of the genetic network T2 (a), T3 (b) and T4 (c), respectively.
(TIF)

**S6 Fig. Motif features in the two-gene system with two different initial cell states.** (a). Schematic diagram of intracellular and intercellular regulatory motifs. (b). With the initial cell state: $g_1^A(t_{start}) = 1, g_1^B(t_{start}) = 0$, relationships between maximal positional information and the proportion of times the corresponding regulatory motif occurs. The proportion of intracellular self-inhibition and intercellular direct-lateral-activation motifs occurrence are negatively correlated with the maximum positional information. While the proportion of intercellular direct-lateral-inhibition motifs occurrence is positively connected with the maximum positional information. (c). With the initial cell state: $g_1^A(t_{start}) = 1, g_1^B(t_{start}) = 1$, relationships between maximal positional information and the proportion of times the corresponding regulatory motif occurs. The proportion of intracellular self-inhibition and intercellular direct-lateral-activation motifs occurrence are negatively correlated with the maximum positional information. While the proportion of intercellular direct-lateral-inhibition motif occurrence is

positively connected with the maximum positional information. (d). With the initial cell state: $g_1^A(t_{start}) = 1, g_1^B(t_{start}) = 1$, the proportion of occurrence of direct lateral inhibition, indirect lateral inhibition, and no lateral inhibition under various higher than 1.0 bit maximum positional information.
(TIF)

**S7 Fig. A modified model in which one endpoint has a fixed special cell that does not divide or go through cell state updates.** (a). The modified model with a fixed special cell is depicted in this diagram. The cell that is fixed at one end does not divide, and the state of the cell remains the same. With the exception of the fixed cell, all other cells have the ability to divide and update their states. Various colors are used to indicate different cell states. (b). An example of the genetic network (T5) that regulates four blocks pattern formation at the 9-cell stage. (c). The time-space pattern of gene expression that is regulated by the genetic network T5 in (b). The system divides four discrete blocks with various cell states at the 9-cell stable state. (d). With the initial cell state: $g_1^A(t_{start}) = 1, g_1^B(t_{start}) = 1$. and fixed cell state: $g_{fixed}^A(t) = 1, g_{fixed}^B(t) = 1$, the proportion of occurrence of direct lateral inhibition, indirect lateral inhibition, and no lateral inhibition under various block numbers. (e). The distribution of motif ratios associated with different block numbers with the initial cell state: $g_1^A(t_{start}) = 1, g_1^B(t_{start}) = 1$ and fixed cell state: $g_{fixed}^A(t) = 1, g_{fixed}^B(t) = 1$. (f). Relationships between maximal positional information and the proportion of times the corresponding regulatory motif occurs with the initial cell state: $g_1^A(t_{start}) = 1, g_1^B(t_{start}) = 1$ and fixed cell state: $g_{fixed}^A(t) = 1, g_{fixed}^B(t) = 1$. The proportion of intercellular direct-lateral-activation motif occurrence is negatively correlated with the maximum positional information. While the proportion of intercellular direct-lateral-inhibition motif occurrence is positively connected with the maximum positional information. (g). The fraction of occurrence of direct lateral inhibition, indirect lateral inhibition, and no lateral inhibition under varied higher than 1.0 bit maximum positional information with the initial cell state: $g_1^A(t_{start}) = 1, g_1^B(t_{start}) = 1$ and fixed cell state: $g_{fixed}^A(t) = 1, g_{fixed}^B(t) = 1$.
(TIF)

**S8 Fig. Impact of state update perturbations and asynchronous cell division on the stability of multicellular states.** (a). When there are only two cells in the system, there are 16 possible initial multicellular states, but only 10 are stable under the regulation of the genetic network T1. The basin size for a stable state is the number of initial multicellular states that converge to it. (b). In a fixed system with four cells, there are 256 possible initial multicellular states, however only 66 of these multicellular states are stable under the regulation of the genetic network T1. (c). When cell cycle length equals seven, the relationship between the probability of cell state update mistakes and pattern accuracy under the regulation of different genetic networks. (d). When the average cell cycle length equals seven, the relationship between the standard deviation ($\sigma$) of cell cycle length and pattern accuracy under the regulation of different genetic networks.
(TIF)

**S9 Fig. Phase portraits in the single-gene system.** (a). The phase portrait of gene A corresponding to Eq *S2*. Colored dashed lines represent zero-solution lines for various external parameters. (b). When the expression of gene A is controlled by Eq *S2*, the relationship between gene A's fixed points and the expression level of gene A in neighboring cells. (c). The phase portrait of gene A corresponding to Eq *S3*. Colored dashed lines represent zero-solution lines for various external parameters. (d). When the expression of gene A is controlled by Eq

*S*3, the relationship between gene A's stable states and the expression level of gene A in neighboring cells. (e). When the expression of gene A is controlled by Eq *S*4, the relationship between gene A's stable states and the expression level of gene A in neighboring cells. (f). Resistance to noise of patterns generated in generation 4 under various regulatory settings in single-gene systems.
(TIF)

**S10 Fig. Phase portraits in the two-gene system corresponding to the genetic network T1.**
(a). When the expression of gene A in the genetic network T1 is described by Eq *S*2, the relationship between gene A's stable states and the expression level of gene A in neighboring cells. (b). When the expression of gene B in the genetic network T1 is described by Eq *S*12, the intracellular gene B's stable state value corresponds to two genes in neighboring cells with different expressions. (c). The spatiotemporal expression patterns regulated by the genetic network T1, and corresponding equations which regulate this pattern formation. (d). When the expression of gene A in the genetic network T1 is described by Eq *S*2, the relationship between gene A's stable states and the expression level of gene A in neighboring cells. (e). When the expression of gene B in the genetic network T1 is described by Eq *S*13, the intracellular gene B's stable state value corresponds to two genes in neighboring cells with different expressions.
(TIF)

**S11 Fig. Phase portraits in the two-gene system corresponding to the genetic network T6.**
(a). Phase portrait of gene B in the Boolean model corresponding to the genetic network T1. When gene B's initial expression level is low, the influence of various variables on its stable state. Different colors represent different stable state values of gene B. (b). Phase portrait of gene B in the Boolean model corresponding to the genetic network T1. When gene B's initial expression level is high, the influence of various variables on its stable state. Different colors represent different stable state values of gene B. (c). Eq *S*16 may regulate the expression of gene B, resulting in a bistable phase portrait of gene B under a set of particular parameters. Colored dashed lines represent zero-solution lines for various external parameters. (d). The spatiotemporal expression patterns regulated by the genetic network T6, and corresponding equations which regulate this pattern formation. (e). When the expression of gene A in the genetic network T6 is described by Eq *S*2, the relationship between gene A's stable states and the expression level of gene A in neighboring cells. (f). The phase portrait of gene B corresponding to Eq *S*16. Colored dashed lines represent zero-solution lines for various external parameters. (g). When the expression of gene B in the genetic network T6 is described by Eq *S*16, the intracellular gene B's stable state value corresponds to two genes in neighboring cells with different expressions
(TIF)

**S11 Fig. Validation of Booleanization by lineage specifiers.** (a). Expression level of lineage specifier tbx-38 separates the two sublineages in the AB to ABa and ABp division. Colors indicate the expression levels of transcription factor tbx-38 in the descendant lineage of ABa (blue) and ABp (green). (b). Heat map for distance between pairs of cells on the booleanized 40-dimensional gene-expression space. (c). Heat map for distance between pairs of cells on the continues 195-dimensional gene-expression space. 186 cells are arranged by their orders on the lineage tree at time 140. (d). Correlation between the distance between cells calculated using a Boolean form of gene expression profile (y-axis) and the distance between cells measured by a continuous form of gene expression profile (x-axis).
(TIF)

## Acknowledgments

We are grateful to Dr. Guoye Guan from Dr. Chao Tang's lab for providing experimental data on the contact connections between cells with us throughout the early embryonic development of *C.elegans*. We greatly appreciate the insightful discussions we had with the members of the Zhiyuan Lab and our other researchers.

## Author Contributions

**Conceptualization:** Zhiyuan Li.

**Data curation:** Lu Zhang, Zhiyuan Li.

**Formal analysis:** Lu Zhang.

**Investigation:** Lu Zhang, Gang Xue, Xiaolin Zhou.

**Methodology:** Zhiyuan Li.

**Supervision:** Jiandong Huang, Zhiyuan Li.

**Visualization:** Lu Zhang, Gang Xue, Xiaolin Zhou.

**Writing – original draft:** Lu Zhang.

**Writing – review & editing:** Jiandong Huang, Zhiyuan Li.

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
