## [Decision Letter · Decision Letter 0]

26 Mar 2024

Dear Dr. Li,

Thank you very much for submitting your manuscript "A mathematical framework for understanding the spontaneous emergence of complexity applicable to growing multicellular systems" for consideration at PLOS Computational Biology. As with all papers reviewed by the journal, your manuscript was reviewed by members of the editorial board and by several independent reviewers. The reviewers appreciated the attention to an important topic. Based on the reviews, we are likely to accept this manuscript for publication, providing that you modify the manuscript according to the review recommendations.

Sincerely,

Attila Csikász-Nagy

Academic Editor

PLOS Computational Biology

Stacey Finley

Section Editor

PLOS Computational Biology

Reviewer's Responses to Questions

**Comments to the Authors:**

Reviewer #1: In the manuscript ‘A mathematical framework for understanding the spontaneous emergence of complexity applicable to growing multicellular systems’, Zhang et al. used an agent-based framework to build one-dimensional models concerning gene expression patterns arising from cell division in early embryos starting from one cell and various types of gene regulatory networks. The models show that sequential cell divisions can create distinct contact patterns among embryonic cells. A specific group of networks containing lateral inhibition can utilize contact patterns to generate stable gene expression patterns with high positional information. They used data in C. elegans development to show that the information of gene expression increases with the contact information driven by cell division. Overall, this is an interesting study for understanding mechanisms underlying pattern formation, but I have some confusions on the approach to quantify positional information. I hope that the authors can clarify some of the key concepts in the revision. I also have a few other suggestions for discussing this work in a broader context.

1. In Ref 16, quantified positional information refers to how precisely a gene expression pattern can convey information of a cell’s position, given the variability of expression. In this work, it is unclear to me how variability can be generated from deterministic simulations shown in Figs 1 and 2. In the illustration of Fig S1, variability across ‘samples’ would reduce positional information, but it is not clear what ‘samples’ here refer to in the simulations. It seems to me that if the simulations are deterministic, then the ‘positional information’ is simply determined by H(state), which, according to Figs 1 and 2, measures the diversity of the states at a particular time point regardless of cells’ positions. I have a hard time understanding the connection between the ‘positional information’ measured in this study and positional precision that one should be concerned about for development.

2. I feel that the lateral inhibition mechanism proposed by the authors have a strong bias towards a ‘salt and pepper’ pattern, and it may reduce positional information in realistic settings. As previously shown (10.15252/msb.20167347), lateral inhibition tends to create a ‘salt and pepper’ pattern where two types of cells are distributed in an intermingled fashion. While this pattern can be beneficial for some tissues, positional information (precision) is presumably low when one looks across multiple samples (embryos), according to the definition in Fig S1. It seems that the authors observed this tendency at the end of the simulation shown in Fig 2d, but they did not comment on this (they also did not comment on the decrease of ‘positional information’ in generation 5, which is another issue to me). This raises concern about the robustness of the pattern. In Fig 1e, for example, if cell 11’s position is taken by a green cell due to cell movement or sporadic gene expression, lateral inhibition would stabilize it, and this would reduce positional information.

3. The authors considered the variability of cell cycle length a form of noise. ‘Noise’ is a misleading term to use here, especially in the context of C. elegans, in which cell divisions are asynchronous and precisely controlled.

4. C. elegans embryos undergo asymmetric cell divisions in which cell polarity is determined before division occurs. Even though this pattern formation mechanism may dependent on cell contact, This is conceptually different from what the authors proposed, i.e. each cell divides symmetrically and then integrates the contact information from multiple neighbors for fate decisions. This biological discrepancy needs to be reconciled.

5. Contact dependent cell differentiation and the role of noise in embryonic patterning were extensively studied previously. In the work of Holmes et al. (10.1371/journal.pcbi.1005320), for example, noise was shown to be beneficial for correcting cell fates for organizing patterns. I suggest that the authors discuss some of these related works.

6. Typo in Line 320: a colon is repeated.

Reviewer #2: This is an interesting theoretical paper, which looks at the emergence of positional information in a proliferating 1D set of cells with juxtacrine signalling. The authors conclude that lateral inhibition motifs are central in systems in which maximal positional information arises.

Overall, I think the paper is interesting and reasonably well written. I have only minor comments:

The authors should make clear the parameter values used in simulations of the continuous model, especially those in figure 4. Also, since results are likely to vary with parameter values, the authors should give an indication of how robust the results are, by performing some kind of parameter sensitivity analysis.

line 300 strip -> stripe

line 310 remove "potentially"

line 312 "lead to" -> "represent"

line 320 remove 2nd comma at end

line 343 a continuous model/ continuous models

line 393 bit -> bits

line 619 this does not make sense, please re-write

line 721 the sum in the brackets is missing g_i^q(t)

line 867 continious -> continuous

The authors should consider if all the supplementary figures are absolutely necessary, as there are very many.

**Have the authors made all data and (if applicable) computational code underlying the findings in their manuscript fully available?**

Reviewer #1: Yes

Reviewer #2: **No: **I don't see the computational code.

PLOS authors have the option to publish the peer review history of their article (what does this mean?). If published, this will include your full peer review and any attached files.

Reviewer #1: No

Reviewer #2: No

Figure Files:

Data Requirements:

Please note that, as a condition of publication, PLOS' data policy requires that you make available all data used to draw the conclusions outlined in your manuscript. Data must be deposited in an appropriate repository, included within the body of the manuscript, or uploaded as supporting information. This includes all numerical values that were used to generate graphs, histograms etc.. For an example in PLOS Biology see here: http://www.plosbiology.org/article/info:doi%2F10.1371%2Fjournal.pbio.1001908#s5.

Reproducibility:

References:

---

## [Decision Letter · Decision Letter 1]

20 May 2024

Dear Dr. Li,

We are pleased to inform you that your manuscript 'A mathematical framework for understanding the spontaneous emergence of complexity applicable to growing multicellular systems' has been provisionally accepted for publication in PLOS Computational Biology.

Best regards,

Attila Csikász-Nagy

Academic Editor

PLOS Computational Biology

Stacey Finley

Section Editor

PLOS Computational Biology

Reviewer's Responses to Questions

**Comments to the Authors:**

Reviewer #1: The authors have addressed all my comments. Congratulations on this wonderful work.

Reviewer #2: The authors have made the changes I suggested and I now think this manuscript is acceptable for publication.

**Have the authors made all data and (if applicable) computational code underlying the findings in their manuscript fully available?**

Reviewer #1: Yes

Reviewer #2: Yes

PLOS authors have the option to publish the peer review history of their article (what does this mean?). If published, this will include your full peer review and any attached files.

Reviewer #1: **Yes: **Tian Hong

Reviewer #2: No

---

## [Editor Report · Acceptance letter]

29 May 2024

PCOMPBIOL-D-24-00204R1 

A mathematical framework for understanding the spontaneous emergence of complexity applicable to growing multicellular systems

Dear Dr Li,

I am pleased to inform you that your manuscript has been formally accepted for publication in PLOS Computational Biology. Your manuscript is now with our production department and you will be notified of the publication date in due course.

With kind regards,

Lilla Horvath
